# Convergence Analysis of Linear Coupling with Inexact Proximal Operator

**Qiang Zhou**[1,2*]   **and**   **Sinno Jialin Pan**[3]

[1]Southeast University, Nanjing 211189, China
[2]Purple Mountain Laboratories, Nanjing 211111, China
[3]Nanyang Technological University, Singapore
zhouqiang@u.nus.edu, sinnopan@ntu.edu.sg

## Abstract

Linear coupling is recently proposed to accelerate first-order algorithms by linking gradient descent and mirror descent together, which is able to achieve an accelerated convergence rate. This work focuses on the convergence analysis of linear coupling for convex composite minimization when a proximal operator cannot be exactly computed. It is of particular interest to study the convergence of linear coupling because it not only achieves the accelerated convergence rate for first-order algorithms but also works for generic norms. We present convergence analysis of linear coupling by allowing the proximal operator to be computed up to a certain precision. Our analysis illustrates that the accelerated convergence rate of linear coupling with an inexact proximal operator can be preserved if the error sequence of the inexact proximal operator decreases in a sufficiently fast rate. More importantly, our analysis leads to better bounds than existing works with inexact proximal operators. Experiment results on several real-world datasets verify our theoretical results.

## 1 INTRODUCTION

In this work, we consider convex composite minimization problems in the form of

$$\min_{\mathbf{x} \in \mathbb{R}^d} f(\mathbf{x}) \stackrel{\text{def}}{=} g(\mathbf{x}) + h(\mathbf{x}), \tag{1}$$

where $g$ is convex or $\mu$-strongly convex, and $L$-smooth, and $h$ is convex but possibly non-smooth [Nesterov, 2005]. Various machine learning problems can be formulated in the form of (1), where $g$ defines a convex loss function for

---

*Most of this work was performed while the first author worked as a postdoc at NTU, Singapore.

training examples, and $h$ regularizes the model to promote a specified structure [Bach et al., 2011, Sra et al., 2012, Jenatton et al., 2011]. For instance, it is well-known that $h(\mathbf{x}) = \|\mathbf{x}\|_1$ can be used to induce a sparse structure for $\mathbf{x}$ [Tibshirani, 1996]. In addition, constrained optimization problems can also be formulated in the form of (1) through reformulation. Specifically, for any convex optimization problem with constraint $\mathbf{x} \in \mathcal{C}$, it can be reformulated as (1) by defining $h(\mathbf{x}) \stackrel{\text{def}}{=} I_{\mathcal{C}}(\mathbf{x})$ as the indicator function of the convex set $\mathcal{C}$, where $I_{\mathcal{C}}(\mathbf{x}) = 0$ if $\mathbf{x} \in \mathcal{C}$ or $I_{\mathcal{C}}(\mathbf{x}) = \infty$ otherwise.

To solve (1), first-order algorithms are the most popular choice due to their simplicity and generality [Sra et al., 2012, Nesterov, 2013]. First-order algorithms generally assume that the first-order gradient of a smooth function can be queried by a black-box in constant time. Therefore, the complexity for solving a smooth and convex function $f$ is measured by the number of times that first-order gradients are queried to produce a sequence $\{\mathbf{x}_k\}_{k=1}^T$ s.t. $f(\mathbf{x}_T) - f(\mathbf{x}^\star) \leq \epsilon$, where $\mathbf{x}^\star$ is the optimal solution. A gradient descent algorithm has $O(L/\epsilon)$ iteration complexity for $L$-smooth convex minimization problems, which can be improved to $O(L/\mu \log(1/\epsilon))$ when the function is $\mu$-strongly convex and $L$-smooth [Nesterov, 2013]. However, these complexities are not optimal, which leaves big space for improvement [Nemirovsky and Yudin, 1983]. In the seminal work [Nesterov, 1983], Nesterov laid the foundation of accelerated gradient methods (AGDs) for convex and $L$-smooth functions (i.e. $h(\mathbf{x}) = 0$ in (1)). Specifically, the method proposed in [Nesterov, 1983] successfully improves the complexities of gradient descent methods to $O(\sqrt{L/\epsilon})$ and $O(\sqrt{L/\mu} \log(1/\epsilon))$ for generally $\mu$-strongly convex and smooth functions, respectively, which are accelerated and optimal for first-order algorithms [Nemirovsky and Yudin, 1983, Nesterov, 2013]. Since then, accelerated first-order algorithms with the optimal convergence rate have been further developed to solve convex composite minimization problems [Tseng, 2008, Lin et al., 2017].

However, the interpretation of the proof of acceleration and

---

*Accepted for the 38$^{th}$ Conference on Uncertainty in Artificial Intelligence (UAI 2022).*

intuitions behind the convergence analysis in Nesterov's AGDs are not clear. Many efforts have been devoted to present a clear interpretation for Nesterov's AGDs, or alternatively to develop new accelerated and interpretable variants with the optimal convergence rate [Bubeck et al., 2015, Su et al., 2014, Krichene et al., 2015, Diakonikolas and Orecchia, 2018]. Recently, Allen Zhu and Orecchia [2017] proved that the optimal convergence rate can be achieved for smooth and convex optimization problems by *linearly coupling* two fundamental first-order algorithms (namely gradient descent and mirror descent). Later, Rodomanov [2016] extended linear coupling to generally convex composite minimization problems.

As a variant of Nesterov's AGDs, linear coupling (LC) elegantly extends AGDs to non-Euclidean norms, which is important for many applications. In [Allen Zhu and Orecchia, 2017, Appendix A.1], several concrete examples have been presented to illustrate the importance of allowing non-Euclidean norms in first-order algorithms. For example, one prefers to use the $\ell_1$ norm instead of the $\ell_2$ norm gradient descent for the saddle point problem $\min_{\mathbf{x} \in \Delta_n} \max_{\mathbf{y} \in \Delta_m} \mathbf{y}^\top \mathbf{A} \mathbf{x}$, where $\Delta_n$ and $\Delta_m$ denotes the unit simplex in $\mathbb{R}^n$ and $\mathbb{R}^m$, respectively, and $\mathbf{A} \in \mathbb{R}^{m \times n}$ with all the entries being in $[-1, 1]$. In this problem, there are two reasons for choosing the $\ell_1$ norm gradient descent as follows.

1) In order to apply the $\ell_2$ norm gradient descent, one needs to further assume the square $\ell_2$ norm of each row of $\mathbf{A}$ is upper bounded by 1. Obviously, it is a stronger condition and harder to satisfy than that of the $\ell_1$ norm.

2) More importantly, even the stronger condition is satisfied, the $\ell_2$ norm also leads to a larger value for $L$, thus the $\ell_1$ norm gradient descent has faster convergence [Nesterov, 2005]. Another example is near-linear time maximum flow in which one need to apply $\ell_\infty$ gradient descent [Kelner et al., 2014].

For convex composite optimization problems (1), the proximal gradient descent [Parikh and Boyd, 2014] performs the following updates:

$$\mathbf{x}_{k+1} = \text{prox}_{\eta_{k+1} h} \big( \mathbf{x}_k - \eta_{k+1} \nabla g(\mathbf{x}_k) \big),$$

where $\text{prox}_{\eta h}(\cdot)$ is the proximal operator [Combettes and Pesquet, 2011] of $\eta h(\mathbf{x})$ defined for any scalar $\eta > 0$ as the unique solution of

$$\text{prox}_{\eta h}(\mathbf{y}) = \operatorname*{argmin}_{\mathbf{x} \in \mathbb{R}^d} \left\{ \eta h(\mathbf{x}) + \frac{1}{2} \|\mathbf{x} - \mathbf{y}\|^2 \right\}. \quad (2)$$

If $h$ is considerably simple (e.g., $h(\mathbf{x}) = \|\mathbf{x}\|_1$), there is an analytical solution for $\mathbf{x}_{k+1}$ [Combettes and Pesquet, 2011]. However, in more general cases, it is challenging to obtain the exact solution of the proximal operator possibly due to the following two reasons.

- First, the proximal operator does not admit an analytical solution. For example, there is no closed form solution for the proximal operator if $h$ is the isotropic total variation regularization [Beck and Teboulle, 2009]. In this case, the proximal operator can only be solved by employing some optimization algorithm up to a certain precision. More details of this example can be found in *Experiments*.

- Second, it may be computationally expensive to obtain the exact solution. For instance, for optimization problems with an $\ell_1$ norm ball constraint (e.g., $h(\mathbf{x}) = I_{\|\mathbf{x}\|_1 \leq r}(\mathbf{x}), r \in \mathbb{R}_+$), the complexity of exactly performing the proximal operator is $O(d \log d)$ [Duchi et al., 2008, Liu and Ye, 2009]. Therefore, it is highly demanding in computation for high-dimensional cases as $d$ is considerably large. We empirically compare the efficiency of exact and inexact proximal operator of the $\ell_1$ norm ball constraint. The result suggests that the inexact proximal operator outperforms the exact counterpart by carefully controlling the error sequences in this application.

In [Allen Zhu and Orecchia, 2017], the objective is assumed to be convex and smooth, i.e., $h(\mathbf{x}) = 0$. Thus, the analysis of linear coupling does not involve the computation of proximal operator. However, it is well-known that many machine learning problems can be formulated as a convex composite minimization problem due to the non-smooth regularization. Therefore, it is of particular interest to characterize the convergence behavior of linear coupling, when the proximal operator is not exactly solved. However, existing analyses [Schmidt et al., 2011] on inexact proximal operators only cover the $\ell_2$ norm case, thus they are not applicable to linear coupling due to non-Euclidean norms. To this end, we present a complete study for the convergence rate of *linear coupling* with inexact proximal operators.

Compared with existing works [Schmidt et al., 2011, Lin et al., 2017, Kulunchakov and Mairal, 2019], our focus is the convergence analysis of linear coupling with inexact proximal operators, which presents new challenges due to the generic Bregman divergence (refer to Definition 3). In particular, the key step for analyzing inexact proximal operators is to bound the subgradient of inexact solution. It is an easy task in the case of squared Euclidean distance as the bounding problem has an analytical solution [Schmidt et al., 2011]. In contrast, it does not admit an analytical form in our case due to the generic Bregman divergence, e.g., the Kullback-Leibler divergence. To address it, we present a relaxation method for the bounding problem so that the subgradient still can be bounded (see Lemma 1). More importantly, our analysis leads to tighter bounds (refer to Remarks 2 and 5 for details) than previous works [Schmidt et al., 2011, Lin et al., 2017, Kulunchakov and Mairal, 2019].

## 2 NOTATION AND PRELIMINARIES

Throughout this paper, vectors and matrices are denoted by lower-case and upper-case boldface characters (e.g., $\mathbf{x}$ and $\mathbf{X}$), respectively. Let $\mathbf{0}$ be a vector or matrix with all its entries equal to 0. For both $\mathbf{x}, \mathbf{y} \in \mathbb{R}^d$, their inner product is denoted by $\langle \mathbf{x}, \mathbf{y} \rangle = \sum_{i=1}^d x_i y_i$. Let $\|\cdot\|$ be a generic norm and its dual norm is denoted by $\|\cdot\|_*$ that is defined as $\|\mathbf{y}\|_* = \sup_{\mathbf{x}} \left\{ \langle \mathbf{x}, \mathbf{y} \rangle \mid \|\mathbf{x}\| \leq 1 \right\}$. Let $\|\cdot\|_1$ and $\|\cdot\|_2$ denote the $\ell_1$ and the $\ell_2$ norm, respectively.

**Definition 1.** *A function $f$ is $L$-smooth w.r.t. $\|\cdot\|$, if*

$$f(\mathbf{y}) \leq f(\mathbf{x}) + \langle \nabla f(\mathbf{x}), \mathbf{y} - \mathbf{x} \rangle + \frac{L}{2}\|\mathbf{y} - \mathbf{x}\|^2, \ \forall \mathbf{x}, \mathbf{y}.$$

**Definition 2.** *A function $f$ is $\mu$-strongly convex w.r.t. $\|\cdot\|$, if*

$$f(\mathbf{y}) \geq f(\mathbf{x}) + \langle \nabla f(\mathbf{x}), \mathbf{y} - \mathbf{x} \rangle + \frac{\mu}{2}\|\mathbf{y} - \mathbf{x}\|^2, \ \forall \mathbf{x}, \mathbf{y}.$$

**Definition 3.** *Let $\psi : \mathcal{Q} \to \mathbb{R}$ be a strictly convex and continuously differentiable function. Then, the Bregman divergence is*

$$V_\psi(\mathbf{y}, \mathbf{x}) \stackrel{\text{def}}{=} \psi(\mathbf{y}) - \psi(\mathbf{x}) - \langle \nabla \psi(\mathbf{x}), \mathbf{y} - \mathbf{x} \rangle, \forall \mathbf{x}, \mathbf{y} \in \mathcal{Q}.$$

Definition 3 implies $V_\psi(\mathbf{x}, \mathbf{x}) = 0$, and $V_\psi(\mathbf{y}, \mathbf{x}) \geq \frac{\rho}{2}\|\mathbf{x} - \mathbf{y}\|^2$ if $\psi$ is $\rho$-strongly convex w.r.t. $\|\cdot\|$. The Bregman divergence includes many well-known examples.

- 1) If $\psi(\mathbf{x}) \stackrel{\text{def}}{=} \frac{1}{2}\|\mathbf{x}\|_2^2$, then $V_\psi(\mathbf{y}, \mathbf{x})$ is the squared Euclidean distance $V_\psi(\mathbf{y}, \mathbf{x}) = \frac{1}{2}\|\mathbf{x} - \mathbf{y}\|_2^2$.

- 2) If $\mathcal{Q} \stackrel{\text{def}}{=} \left\{ \mathbf{x} \in \mathbb{R}_+^d \mid \sum_i x_i = 1 \right\}$ and $\psi(\mathbf{x}) \stackrel{\text{def}}{=} \sum_i x_i \log x_i$, then $V_\psi(\mathbf{y}, \mathbf{x})$ becomes the Kullback-Leibler divergence $V_\psi(\mathbf{y}, \mathbf{x}) = \sum_i y_i \log(\frac{y_i}{x_i})$ between two probability distributions $\mathbf{x}$ and $\mathbf{y}$. In particular, $\psi(\mathbf{x})$ in this case is 1-strongly convex w.r.t. $\|\cdot\|_1$ that leads to $V_\psi(\mathbf{y}, \mathbf{x}) \geq \|\mathbf{x} - \mathbf{y}\|_1^2$. In this case, one needs to employ optimization methods that are applicable for the $\ell_1$ norm. Therefore, this example illustrates the importance of convergence analysis for linear coupling with inexact proximal operators since it works for a generic norm.

**Definition 4.** *For function $f(\mathbf{x})$, its convex conjugate is defined as*

$$f^*(\mathbf{y}) \stackrel{\text{def}}{=} \sup_{\mathbf{x}} \left\{ \langle \mathbf{x}, \mathbf{y} \rangle - f(\mathbf{x}) \right\}.$$

## 3 LINEAR COUPLING WITH INEXACT PROXIMAL OPERATOR

For smooth and convex functions, the accelerated convergence rate can be obtained by two non-accelerated algorithms: linearly coupling gradient descent and mirror descent [Allen Zhu and Orecchia, 2017]. In fact, it can also

---

**Algorithm 1** LC with Inexact Proximal Operators

1: **Input**: $\mathbf{x}_0, \alpha_0, \mu, L$
2: **Initialization**: $\mathbf{y}_0 \leftarrow \mathbf{x}_0, \mathbf{z}_0 \leftarrow \mathbf{x}_0$
3: **for** $k = 0$ **to** $T-1$ **do**
4:     **if** $\mu = 0$ **then**
5:         Set $\eta_{k+1} \leftarrow \frac{k+2}{2L}$ and $\tau_k \leftarrow \frac{1}{L\eta_{k+1}}$
6:         Set $\mathbf{w}_{k+1} \leftarrow \mathbf{z}_k$
7:     **else**
8:         Set $\eta_{k+1} \leftarrow \frac{1}{L\alpha_{k+1}}$ and $\tau_k \leftarrow \frac{L\alpha_{k+1} - \mu}{L - \mu}$ where $\alpha_{k+1}$ is obtained via $\alpha_{k+1}^2 = (1 - \alpha_{k+1})\alpha_k^2 + \frac{\mu}{L}\alpha_{k+1}$
9:         Set $\mathbf{w}_{k+1} \leftarrow \frac{\tau_k}{\alpha_{k+1}}\mathbf{z}_k + \left(1 - \frac{\tau_k}{\alpha_{k+1}}\right)\mathbf{y}_k$
10:     **end if**
11:     $\mathbf{x}_{k+1} \leftarrow \tau_k \mathbf{z}_k + (1 - \tau_k)\mathbf{y}_k$
12:     Find a $\xi_{k+1}$-suboptimal solution $\mathbf{y}_{k+1}$ for (3)
13:     Find a $\xi_{k+1}$-suboptimal solution $\mathbf{z}_{k+1}$ for (4)
14: **end for**
15: **Output**: $\mathbf{y}_T$

---

be extended to solve convex composite minimization problem (1) [Rodomanov, 2016]. We assume that $g$ is convex ($\mu = 0$) or $\mu$-strongly convex ($\mu > 0$), and $L$-smooth w.r.t. $\|\cdot\|$. Here, we present the extension of linear coupling (LC) for (1) when $g$ is either convex ($\mu = 0$) or strongly convex ($\mu > 0$) and summarize the high-level idea in Algorithm 1. Missing proofs can be found in Appendix.

Let $\mathbf{y}_k$ and $\mathbf{z}_k$ be the outputs of gradient descent and mirror descent in the $(k-1)$-th iteration, respectively. The key idea of linear coupling is to combine $\mathbf{y}_k$ and $\mathbf{z}_k$ together by a linear coupling rate $\tau_k$ as the starting point for the next iteration such that the accelerated convergence rate can be achieved.

Define $\mathbf{x}_{k+1} \stackrel{\text{def}}{=} \tau_k \mathbf{z}_k + (1 - \tau_k)\mathbf{y}_k$. In linear coupling, the gradient descent performs the following update:

$$\mathbf{y}_{k+1} = \operatorname*{argmin}_{\mathbf{y}} \widetilde{Q}_{k+1}(\mathbf{y}; \mathbf{x}_{k+1}), \qquad (3)$$

where $\widetilde{Q}_{k+1}(\mathbf{y}; \mathbf{x}_{k+1})$ is define as

$$\widetilde{Q}_{k+1}(\mathbf{y}; \mathbf{x}_{k+1}) = \langle \nabla g(\mathbf{x}_{k+1}), \mathbf{y} \rangle + h(\mathbf{y}) + \frac{L}{2}\|\mathbf{y} - \mathbf{x}_{k+1}\|^2.$$

Define $\eta_{k+1}, \tau_k$ and $\mathbf{w}_{k+1}$ as in Algorithm 1, the mirror descent performs the following update:

$$\mathbf{z}_{k+1} = \operatorname*{argmin}_{\mathbf{x}} \widehat{Q}_{k+1}(\mathbf{z}; \mathbf{w}_{k+1}), \qquad (4)$$

where

$$\widehat{Q}_{k+1}(\mathbf{z}; \mathbf{w}_{k+1}) \stackrel{\text{def}}{=} \langle \nabla g(\mathbf{x}_{k+1}), \mathbf{z} \rangle + h(\mathbf{z}) + \frac{V_\psi(\mathbf{z}, \mathbf{w}_{k+1})}{\eta_{k+1}}.$$

Unlike standard gradient descent and mirror descent, the linear coupling takes the gradient at $\mathbf{x}_{k+1}$ instead of $\mathbf{y}_k$ or $\mathbf{z}_k$ to obtain $\mathbf{y}_{k+1}$ and $\mathbf{z}_{k+1}$. In this way, the gradient

descent and mirror descent are coupled together to solve (1), which is able to achieve the optimal convergence rate [Allen Zhu and Orecchia, 2017].

## 3.1 THE $\xi_{k+1}$-SUBOPTIMAL SOLUTION

The proximal operator is involved in both (3) and (4) due to the non-smooth function $h$. It is worth noting that the proximal distance in both (3) and (4) are defined based a generic norm $\|\cdot\|$ instead of $\|\cdot\|_2$. Specifically, they are the squared norm $\|\cdot\|^2$ and the general Bregman divergence $V_\psi(\cdot,\cdot)$, respectively. In contrast, existing works mainly focus on the case of the squared Euclidean distance (the $\ell_2$ norm). Therefore, (3) and (4) are more challenging to solve than existing works. In other words, it is often the case that (3) and (4) can only be solved up to a certain precision. Therefore, it is critical to study the convergence rate of Algorithm 1 by allowing the proximal operator of the form (3) and (4) to be solved approximately.

For the case of inexact proximal operators, we introduce $\xi_{k+1}$-suboptimal solution to analyze the convergence rate of Algorithm 1. We assume that $h$ is equipped with an oracle such that the proximal operator can be computed up to a certain precision. Specifically, given a non-negative $\xi_{k+1}$, the oracle is able to produce $\mathbf{y}_{k+1}$ and $\mathbf{z}_{k+1}$ such that

$$\widetilde{Q}_{k+1}(\mathbf{y}_{k+1};\mathbf{x}_{k+1}) - \min_{\mathbf{y}} \widetilde{Q}_{k+1}(\mathbf{y};\mathbf{x}_{k+1}) \leq \xi_{k+1}, \quad (5)$$

$$\widehat{Q}_{k+1}(\mathbf{z}_{k+1};\mathbf{w}_{k+1}) - \min_{\mathbf{z}} \widehat{Q}_{k+1}(\mathbf{z};\mathbf{w}_{k+1}) \leq \xi_{k+1}. \quad (6)$$

If $\xi_{k+1} = 0$, it implies that the proximal operator is exactly solved for both (3) and (4). Otherwise, it means that the proximal operator is solved up to a certain precision controlled by $\xi_{k+1}$. Thus, $\mathbf{y}_{k+1}$ and $\mathbf{z}_{k+1}$ are referred to as $\xi_{k+1}$-suboptimal solutions to (3) and (4), respectively.

Note that the inexactness criteria (5) and (6) are same as those used in [Schmidt et al., 2011, Lin et al., 2017, Kulunchakov and Mairal, 2019]. Whereas such a type of criteria has limitations, it remains the most standard one for convergence analysis with inexact proximal operator.

Our analysis allows that the sub-problems (3) and (4) do not admit closed-form solutions Therefore one needs to compute an approximate solution up to a certain accuracy with some iterative algorithm $\mathcal{M}$. Note that both the sub-problems (3) and (4) are strongly convex even the objective $f(\mathbf{x})$ is not strongly convex. The strong convexity allows us to efficiently obtain a $\xi_{k+1}$-suboptimal solution via $\mathcal{M}$ with a linear convergence rate [Lin et al., 2017, Kulunchakov and Mairal, 2019].

## 4 CONVERGENCE ANALYSIS

In this section, we present the convergence analysis for Algorithm 1 when applying it to solve (1). Specifically, we first present properties of a suboptimal solution in Section 4.1. Then, in Sections 4.2 and 4.3, we present specific convergence results of Algorithm 1 for $\mu = 0$ and $\mu > 0$, in Theorems 1 and 2, respectively. For convenience, we also assume that $\psi(\mathbf{x})$ is 1-strongly convex and $\rho$-smooth w.r.t. $\|\cdot\|$. By introducing the Bregman divergence, our analysis can include more cases than existing works. In other words, it can recover many general cases. For example, the counterpart considers the squared $\ell_2$-norm that can be easily obtained by setting $\psi(\mathbf{x}) = \frac{1}{2}\|\mathbf{x}\|_2^2$ where $\psi(\mathbf{x})$ is 1-strongly convex and 1-smooth w.r.t. the $\ell_2$-norm.

## 4.1 PROPERTIES OF SUBOPTIMAL SOLUTION

For convergence analysis with inexact proximal operators, the key is to bound the solution of inexact proximal operator and the $\xi$-subgradient of the inexact solution. Thus, we first introduce the definition of $\xi$-subgradient, which is a generalization of subgradient.

**Definition 5.** *[Bertsekas et al., 2003] For convex function* $f : \mathbb{R}^d \to \mathbb{R}$ *and a non-negative scalar* $\xi$, $\partial_\xi f(\mathbf{x})$ *is the* $\xi$-*subgradient of* $f$ *at* $\mathbf{x}$ *if it holds that,* $\forall \mathbf{v} \in \partial_\xi f(\mathbf{x})$,

$$f(\mathbf{y}) \geq f(\mathbf{x}) + \langle \mathbf{v}, \mathbf{y} - \mathbf{x} \rangle - \xi, \forall \mathbf{y} \in \mathbb{R}^d. \quad (7)$$

Definition 5 implies that $\mathbf{0}$ is a $\xi$-subgradient of $f$ at $\mathbf{x}$ if $\mathbf{x}$ is a $\xi$-suboptimal solution of $f$. In the case of the $\ell_2$ norm, Schmidt et al. [2011] showed that $\mathbf{v}$ in Definition 5 can be easily bounded as the squared Euclidean distance in (2) which leads to an analytical form for $\mathbf{v}$. In contrast, our case is more challenging as it involves $(\nabla\psi)^{-1}$ which generally does not has an analytical form for generic Bregman divergence. To address this problem, we present a relaxation method by exploiting the strongly convexity of $\psi$ so that it still admits an analytical form. We define

$$Q_{k+1}(\mathbf{z};\mathbf{w}_{k+1}) \overset{\text{def}}{=} \langle \nabla g(\mathbf{x}_{k+1}), \mathbf{z} - \mathbf{w}_{k+1} \rangle + h(\mathbf{z}).$$

If $\mathbf{z}_{k+1}$ is $\xi_{k+1}$-suboptimal to (4), Lemma 1 provides a $\xi_{k+1}$-subgradient for $Q_{k+1}(\cdot;\mathbf{w}_{k+1})$ at $\mathbf{z}_{k+1}$.

**Lemma 1.** *For* $\forall k \geq 0$, *if* $\mathbf{z}_{k+1}$ *is a* $\xi_{k+1}$-*suboptimal solution to (4) in the sense of (6), then there exists* $\boldsymbol{\beta}_{k+1}$ *with* $\|\boldsymbol{\beta}_{k+1}\|_*^2 \leq 2\rho\xi_{k+1}/\eta_{k+1}$ *such that*

$$\frac{\nabla\psi(\mathbf{w}_{k+1}) - \nabla\psi(\mathbf{z}_{k+1})}{\eta_{k+1}} - \boldsymbol{\beta}_{k+1} \in \partial_{\xi_{k+1}} Q_{k+1}(\mathbf{z}_{k+1};\mathbf{w}_{k+1}).$$

The proof of Lemma 1 is given in Appendix A.1. By using Lemma 1, the following lemma enables us to bound the intermediate results of mirror descent with the inexact proximal operator.

**Lemma 2.** *Under the same setting as in Lemma 1, then there exists* $\boldsymbol{\beta}_{k+1}$ *with* $\|\boldsymbol{\beta}_{k+1}\|_*^2 \leq 2\rho\xi_{k+1}/\eta_{k+1}$ *such that*

$$\widehat{Q}_{k+1}(\mathbf{z}_{k+1};\mathbf{w}_{k+1}) + V_\psi(\mathbf{u},\mathbf{z}_{k+1})/\eta_{k+1} - \xi_{k+1}$$
$$\leq \widehat{Q}_{k+1}(\mathbf{u};\mathbf{w}_{k+1}) + \langle \boldsymbol{\beta}_{k+1}, \mathbf{u} - \mathbf{z}_{k+1} \rangle, \forall \mathbf{u}, k \geq 0.$$

The proof of Lemma 2 is given in Appendix A.2.

## 4.2 CONVERGENCE RATES OF CONVEX $g$

We first focus on the case when $g$ is convex. The next lemma derives a characteristic inequality for a specific Lyapunov function for inexact linear coupling by considering the inexactness of $\mathbf{y}_{k+1}$ and $\mathbf{z}_{k+1}$.

**Lemma 3.** *Under the same setting as in Lemma 1, for any $k \geq 0$, if $g$ is convex and $\tau_k = 1/L\eta_{k+1}$,*

$$
\frac{1}{\tau_k^2}\left(f(\mathbf{y}_{k+1}) - f(\mathbf{x}^\star)\right) + LV_\psi(\mathbf{x}^\star, \mathbf{z}_{k+1})
$$
$$
\leq \frac{1 - \tau_k}{\tau_k^2}\left(f(\mathbf{y}_k) - f(\mathbf{x}^\star)\right) + LV_\psi(\mathbf{x}^\star, \mathbf{z}_k)
$$
$$
+ \sqrt{\frac{2\rho L\xi_{k+1}}{\tau_k}}\|\mathbf{x}^\star - \mathbf{z}_{k+1}\| + \frac{1 + \tau_k}{\tau_k^2}\xi_{k+1}. \quad (8)
$$

The proof of Lemma 3 is given in Appendix B.1. From the Lyapunov function, we obtain a general convergence result for linear coupling with inexact proximal operator.

**Theorem 1.** *Under the same setting as in Lemma 1, if $g$ is convex, $\eta_{k+1} = (k + 2)/2L$ and $\tau_k = 1/L\eta_{k+1}, \forall k \geq 0$, then $\forall T \geq 1$:*

$$
f(\mathbf{y}_T) - f(\mathbf{x}^\star) \leq \frac{6\left(LV_\psi(\mathbf{x}^\star, \mathbf{x}_0) + \widetilde{E}_T + \widehat{E}_T\right)}{(T + 1)^2}, \quad (9)
$$

*where* $\widetilde{E}_T \overset{\text{def}}{=} \sum_{k=1}^T (k + 2)^2 \xi_k$ *and* $\widehat{E}_T \overset{\text{def}}{=} \left(\sum_{k=1}^T \sqrt{2\rho(k+1)\xi_k}\right)^2$.

The proof of Theorem 1 is given in Appendix B.2.

**Remark 1.** *If the proximal operator is exact (i.e., $\xi_k = 0, \forall k \geq 1$), it leads to $\widetilde{E}_T = 0$ and $\widehat{E}_T = 0$. Then, Theorem 1 recoveries the accelerated complexity $O(\sqrt{L/\epsilon})$ for convex objectives [Nesterov, 2013].*

**Corollary 1.** *Consider the same setting as Theorem 1, if $\xi_k \leq \xi$ for all $k \geq 1$, then $\forall T \geq 1$:*

$$
f(\mathbf{y}_T) - f(\mathbf{x}^\star) \leq \frac{6LV_\psi(\mathbf{x}^\star, \mathbf{x}_0)}{(T + 1)^2} + (\theta_1 T + \theta_2)\xi. \quad (10)
$$

*where $\theta_1 = (6 + 16\rho)$ and $\theta_2 = 15 + 32\rho$.*

**Remark 2.** *Under the same setting as Corollary 1, based on [Schmidt et al., 2011, Proposition 4], we have*

$$
f(\mathbf{y}_T) - f(\mathbf{x}^\star) \leq \frac{6L\|\mathbf{x}_0 - \mathbf{x}^\star\|^2}{(T + 1)^2} + \left(12T^2 + 4T + 2\right)\xi. \quad (11)
$$

*Based on [Lin et al., 2017, Theorem 3], we have*

$$
f(\mathbf{y}_T) - f(\mathbf{x}^\star) \leq \frac{4L\|\mathbf{x}_0 - \mathbf{x}^\star\|^2}{(T + 1)^2} + \frac{9}{2}(T + 2)^2\xi. \quad (12)
$$

*Based on [Kulunchakov and Mairal, 2019, Proposition 4], we have*

$$
f(\mathbf{y}_T) - f(\mathbf{x}^\star) \leq \frac{2e^{1+\gamma}L\|\mathbf{x}_0 - \mathbf{x}^\star\|^2}{(T + 1)^2} + \frac{e^{1+\gamma}}{2\gamma}(T + 2)^2\xi. \quad (13)
$$

*Comparing (10) with (11), (12) and (13), our analysis achieves better bound than existing works [Schmidt et al., 2011, Lin et al., 2017, Kulunchakov and Mairal, 2019]. Specifically, our bound on the inexact proximal operator $\xi$ increases as $O(T\xi)$ while it is $O(T^2\xi)$ for existing works.*

Theorem 1 suggests that the inexact proximal operator leads to error accumulation in the convergence result. To preserve the accelerated rate $O(\frac{1}{T^2})$, Theorem 1 implies the error sequence should decrease to 0 at a sufficiently fast rate.

**Corollary 2.** *Under the same setting as Theorem 1, for any $\delta > 0$, if $\xi_k$ is chosen as $\xi_k \overset{\text{def}}{=} \frac{f(\mathbf{x}_0) - f(\mathbf{x}^\star)}{(k+2)^{3+\delta}}$, then $\forall T \geq 1$,*

$$
f(\mathbf{y}_T) - f(\mathbf{x}^\star) \leq \frac{LV_\psi(\mathbf{x}^\star, \mathbf{x}_0)}{(T + 1)^2}\left(1 + \frac{2}{\delta} + \frac{8\rho}{\delta^2}\right). \quad (14)
$$

**Remark 3.** *To preserve the accelerated rate $O(1/k^2)$ for convex objectives, existing works [Schmidt et al., 2011, Lin et al., 2017, Kulunchakov and Mairal, 2019] requires the error sequences decreases at the rate of $O(1/k^{4+\delta})$. In contrast, our analysis suggests $O(1/k^{3+\delta})$ is sufficient. This is consistent with our tighter bound shown in Corollary 1.*

As observed in (14), the objective value converges faster with more accurate proximal operator (i.e., a larger value for $\delta$). However, a larger $\delta$ also requires more computation time for each iteration.

**Comparison with Alternating Direction Method of Multipliers (ADMM)** Besides the accelerated proximal method, ADMM is another popular method for solving (1) due to its simplicity and applicability to broad applications [Boyd et al., 2011]. It also allows inexact minimization of subproblem to some extent. However, it is well-known that ADMM converges at the rate of $O(1/T)$ for convex objectives [He and Yuan, 2012]. In contrast, the convergence rate of accelerated proximal method is $O(1/T^2)$, which is more desirable for large-scale machine learning problems.

As discussed in [Boyd et al., 2011], ADMM will converge even the when the subproblems of each iteration are not solved exactly, as long as the approximate solutions satisfy certain suboptimality measures. In other words, ADMM also suffers from error accumulation if the sub-problems cannot be exactly solved. For example, the proximal operator does not admit a closed-form solution. We note our inexactness conditions (5) and (6) are absolute criteria which are same as the those used in [Schmidt et al., 2011, Lin et al., 2017, Kulunchakov and Mairal, 2019]. In contrast, existing works on convergence analysis of ADMM are mainly based on a relative error accuracy [He et al., 2002, Eckstein and Yao,

2018, Alves et al., 2020] that is generally a stronger inexactness condition than ours. If ADMM takes the inexactness criterion as ours, under the same setting as Corollary 1, one can show that the convergence rate of inexact ADMM is

$$f(\mathbf{y}_T) - f(\mathbf{x}^\star) \leq \frac{\nu\|\mathbf{x} - \mathbf{x}^\star\|^2}{T} + \gamma T\xi,$$

where $\nu$ and $\gamma$ are some constants. Comparing it with (10), we can observe that both ADMM and our result have $O(T\xi)$ error accumulation. However, our result achieves an $O(1/T^2)$ convergence rate while the rate is only $O(1/T)$ for ADMM.

### 4.3 CONVERGENCE RATES OF STRONGLY CONVEX $g$

In this section, we present the convergence result of linear coupling with inexact proximal operators for convex $g$. Next, we present the convergence result for strongly convex $g$. In this setting we assume $\|\cdot\| = \|\cdot\|_2$ that follows the customary [Nesterov, 2013] of convergence analysis of optimization algorithms for strongly convex objectives. It is mainly used to simplify the proof of convergence analysis. Note that by far, whether the same properties obtained by the $\ell_2$-norm can be generalized to other norms is still an open question. However, even for the $\ell_2$-norm, as we shall see in Remark 4, our convergence rate is still better than previously works.

We first introduce the analogue of Lemma 3 for strongly convex $g$. By considering the inexactness of $\mathbf{y}_{k+1}$ and $\mathbf{z}_{k+1}$, the next lemma derives a characteristic inequality for a specific Lyapunov function for strongly convex objectives.

**Lemma 4.** *Under the same setting as Lemma 1, if $g$ is strongly convex, $\tau_k = \frac{L\alpha_{k+1}-\mu}{L-\mu}$ and $\eta_{k+1} = \frac{1}{L\alpha_{k+1}}$, then $\forall k \geq 0$,*

$$f(\mathbf{y}_{k+1}) - f(\mathbf{x}^\star) + L\alpha_{k+1}^2 V_\psi(\mathbf{x}^\star, \mathbf{z}_{k+1})$$
$$\leq \left(1 - \alpha_{k+1}\right)\left(f(\mathbf{y}_k) - f(\mathbf{x}^\star) + L\alpha_k^2 V_\psi(\mathbf{x}^\star, \mathbf{z}_k)\right)$$
$$+ \sqrt{2\rho L\alpha_{k+1}^3 \xi_{k+1}}\|\mathbf{x}^\star - \mathbf{z}_{k+1}\| + \left(1 + \alpha_{k+1}\right)\xi_{k+1}.$$
(15)

The proof of Lemma 4 is given in Appendix A.8. From the Lyapunov function, we obtain a general convergence result for linear coupling with inexact proximal operators.

We define $\Delta_k \stackrel{\text{def}}{=} f(\mathbf{y}_k) - f(\mathbf{x}^\star) + L\alpha_k^2 V_\psi\left(\mathbf{x}^\star, \mathbf{z}_k\right), \forall k \geq 0$. Theorem 2 presents the convergence result of linear coupling with inexact proximal operators for strongly convex $g$.

**Theorem 2.** *Under the same setting as Lemma 1, if $g$ is strongly convex, $\tau_k = \frac{L\alpha_{k+1}-\mu}{L-\mu}$ and $\eta_{k+1} = \frac{1}{L\alpha_{k+1}}$, then $\forall T \geq 1$:*

$$\Delta_T \leq \Gamma_T\left(\Delta_0 + \sum_{k=1}^T \frac{\sqrt{2\rho L\alpha_k^3\xi_k}\|\mathbf{x}^\star - \mathbf{z}_k\| + 2\xi_k}{\Gamma_k}\right), \quad (16)$$

*where $\Gamma_k \stackrel{\text{def}}{=} \prod_{i=1}^k (1 - \alpha_k)$. If $\alpha_0 = \sqrt{\frac{\mu}{L}}$, then $\forall T \geq 1$:*

$$f(\mathbf{y}_T) - f(\mathbf{x}^\star) \leq \left(1 - \sqrt{\frac{\mu}{L}}\right)^T\left(3\widetilde{\Delta}_0 + \widetilde{R}_T + \widehat{R}_T\right), \quad (17)$$

*where $\widetilde{\Delta}_0 \stackrel{\text{def}}{=} f(\mathbf{x}_0) - f(\mathbf{x}^\star), \widetilde{R}_T \stackrel{\text{def}}{=} 3\sum_{k=1}^T \left(1 - \sqrt{\frac{\mu}{L}}\right)^{-k}\xi_k$ and $\widehat{R}_T \stackrel{\text{def}}{=} 6\rho\sqrt{\frac{\mu}{L}}\left(\sum_{k=1}^T \left(1 - \sqrt{\frac{\mu}{L}}\right)^{-k/2}\sqrt{\xi_k}\right)^2.$*

The proof of Theorem 2 is given in Appendix A.9.

**Remark 4.** *If the proximal operator is exact (i.e., $\xi_k = 0, \forall k \geq 1$), it leads to $\widetilde{R}_T = 0$ and $\widehat{R}_T = 0$. Then, (17) recoveries the accelerated complexity $O(\sqrt{L/\mu}\log(1/\epsilon))$ for $\mu$-strongly convex objectives [Nesterov, 2013].*

**Corollary 3.** *Consider the same setting as Theorem 2, if $\xi_k \leq \xi$ for all $k \geq 1$, then $\forall T \geq 1$:*

$$f(\mathbf{y}_T) - f(\mathbf{x}^\star) \leq 3\left(1 - \sqrt{\frac{\mu}{L}}\right)^T\widetilde{\Delta}_0 + (3 + 24\rho)\sqrt{\frac{L}{\mu}}\xi. \quad (18)$$

**Remark 5.** *Under the same setting as Corollary 3, based on [Schmidt et al., 2011, Proposition 4], we have*

$$f(\mathbf{y}_T) - f(\mathbf{x}^\star) \leq 4\left(1 - \sqrt{\frac{\mu}{L}}\right)^T\widetilde{\Delta}_0 + \left(\frac{64L^2}{\mu^2} + 4\sqrt{\frac{\mu}{L}}\right)\xi. \quad (19)$$

*Based on [Lin et al., 2017, Theorem 3], we have*

$$f(\mathbf{y}_T) - f(\mathbf{x}^\star) \leq 4\left(1 - \sqrt{\frac{\mu}{L}}\right)^T\widetilde{\Delta}_0 + \frac{72L}{\mu}\xi. \quad (20)$$

*Based on [Kulunchakov and Mairal, 2019, Proposition 4], we have*

$$f(\mathbf{y}_T) - f(\mathbf{x}^\star) \leq 2\left(1 - \frac{1}{2}\sqrt{\frac{\mu}{L}}\right)^T\widetilde{\Delta}_0 + \frac{8L}{\mu}\xi. \quad (21)$$

*Comparing (18) with (19), (20) and (21), our analysis achieves better bound than existing works. Specifically, our error bound on $\xi$ is $O(\sqrt{\frac{L}{\mu}}\xi)$, while that are $O(\frac{L^2}{\mu^2}\xi)$ for [Schmidt et al., 2011] and $O(\frac{L}{\mu}\xi)$ for [Lin et al., 2017, Kulunchakov and Mairal, 2019], respectively.*

To preserve accelerated rate for Algorithm 1, Theorem 2 implies that the error sequence $\{\xi_k\}_{k\geq 1}$ needs to decrease to 0 at a linear rate.

**Corollary 4.** *Under the same setting as in Theorem 2, for any $\vartheta \in (0, \sqrt{\mu/L})$, if $\xi_k$ is chosen as $\xi_k \leq \frac{1}{1+2\rho}\widetilde{\Delta}_0(1 - \vartheta)^k$, then $\forall T \geq 1$*

$$f(\mathbf{y}_T) - f(\mathbf{x}^\star) \leq (1 - \vartheta)^{T+1}\frac{12\widetilde{\Delta}_0}{(\sqrt{\mu/L} - \vartheta)^2}. \quad (22)$$

Similar to the conclusion for convex $g$, the objective with strongly convex $g$ has a faster convergence speed when the error of inexact proximal operators decreases at a faster rate (i.e., a larger value for $\rho$).

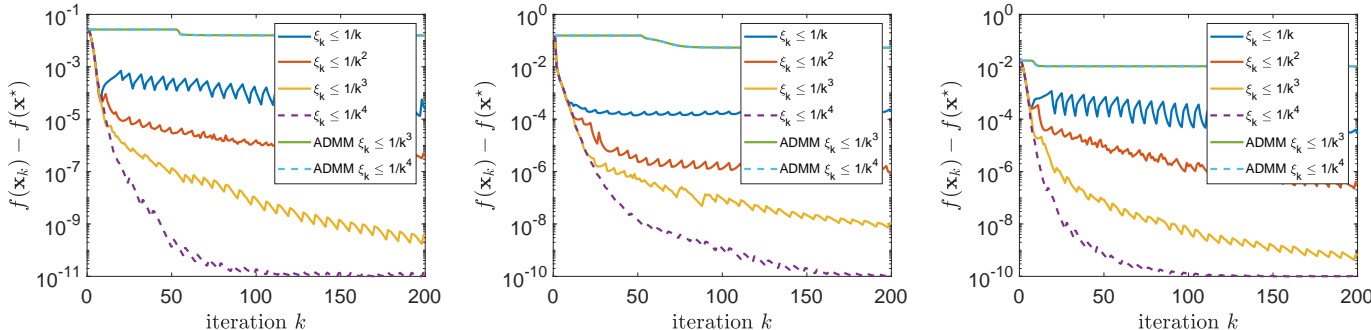

Figure 1: Results of linear coupling and ADMM with inexact proximal operator for CUR-like factorization. Objective function values v.s. number of iterations for different qualities of approximate solution of the proximal operator. From left to right: ala, secom and mushroom. Better viewed on the screen with zooming-in as the difference of ADMM with $\xi_k \leq O(1/k^3)$ and $\xi_k \leq O(1/k^4)$ becomes insignificant compared to the results of linear coupling.

## 5 EXPERIMENTS

In this section, we conduct two experiments to verify our theoretical results.

### 5.1 CUR-LIKE FACTORIZATION

We first apply Algorithm 1 to solve the CUR-like factorization optimization problem [Mairal et al., 2011]. For a given matrix $\mathbf{D} \in \mathbb{R}^{m \times n}$, the CUR-like factorization aims to approximate $\mathbf{D}$ by a matrix $\mathbf{X}$ with sparse rows and sparse columns.

$$\min_{\mathbf{X} \in \mathbb{R}^{m \times n}} \frac{1}{2} \|\mathbf{D}\mathbf{X}\mathbf{D} - \mathbf{D}\|_F^2 + \lambda_1 \sum_{i=1}^{m} \|\mathbf{X}_{i,\cdot}\|_2 + \lambda_2 \sum_{j=1}^{n} \|\mathbf{X}_{\cdot,j}\|_2,$$

where $\mathbf{X}_{i\cdot}$ and $\mathbf{X}_{\cdot,j}$ denote the $i$-th row and the $j$-th column of $\mathbf{X}$, respectively. The last two terms impose the $\ell_{2,1}$ norms for both row and columns of $\mathbf{X}$, that yields both sparse rows and columns. However, the proximal operator of this regularizer does not admit an analytical solution. There is even no iterative algorithm that can exactly compute the proximity operator.

Following [Schmidt et al., 2011], we approximately compute the proximal operator by a block coordinate descent (BCD) algorithm that is presented by Jenatton et al. [2011] to efficiently obtain an approximate solution of the proximal operator. The BCD alternates between computing the proximal operator with respect to the rows and to the columns. At each iteration, we employ the BCD to solve (3) and (4) until (5) and (6) are satisfied that means both $\mathbf{y}_k$ and $\mathbf{z}_k$ are $\xi_k$-suboptimal solutions.

Suggested by Corollary 2, we consider a decreased error sequences $\{\xi_k\}_{k \geq 1}$ where $\xi_k \leq 1/k^\alpha$ and the value of $\alpha$ are set to $\alpha = 1, 2, 3, 4$ in our experiments. As discussed before, the accelerated convergence rate of linear coupling can be preserved if $\alpha > 3$.

We perform experiments on four data sets[1]: mushroom, secom, ala and musk. We set $\lambda_1 = 0.01$ and $\lambda_2 = 0.01$ for all four datasets. Rather than assume the Lipschitz constant $L$ is known, we estimate it by line search [Nesterov, 2013]. Specifically, we initialize the value of $L$ as $L = 0.5$ and double it if the following inequality is not satisfied

$$g(\mathbf{y}_k) \leq g(\mathbf{x}_k) + \langle \nabla g(\mathbf{x}_k), \mathbf{y}_k - \mathbf{x}_k \rangle + \frac{L}{2} \|\mathbf{y}_k - \mathbf{x}_k\|^2.$$

In our experiments, we observed that this strategy always performs better than a fixed but conservative value for $L$.

To demonstrate the accelerated convergence rate of linear coupling, we compare it with ADMM. It is well-know that the performance of ADMM is highly dependent on the choice of penalty parameter $\varrho$. To show the best performance of ADMM, we perform a grid search to find the best value of $\varrho$ and fixed it for all iterations.

Fig. 1 shows the objective function values versus the number of iterations of inexact linear coupling on the three data sets: ala, secom and mushroom. For linear coupling, the choice of $\xi_k \leq 1/k^4$ achieves the fastest convergence rate according to our analysis (refer to Corollary 2), provides the best empirical performance across all three data sets. However, as the iteration goes, the performance gap between the choices of $\xi_k \leq 1/k^3$ and $\xi_k \leq 1/k^4$ becomes smaller. This is consistent with our theoretical results that the error sequences only need to decrease faster than $O(1/k^3)$ instead of $O(1/k^4)$ [Schmidt et al., 2011, Lin et al., 2017, Kulunchakov and Mairal, 2019].

Furthermore, as observed from Fig. 1, the linear coupling clearly outperforms the ADMM on this task. Here, we show the results of ADMM with $\xi_k \leq 1/k^3$ and $\xi_k \leq 1/k^4$. In fact, we observed the result of ADMM when $\alpha \geq 2$ are very close to each other, that is consistent with the theoretical

---

[1]The datasets can be downloaded at https://archive.ics.uci.edu/ml/datasets.php.

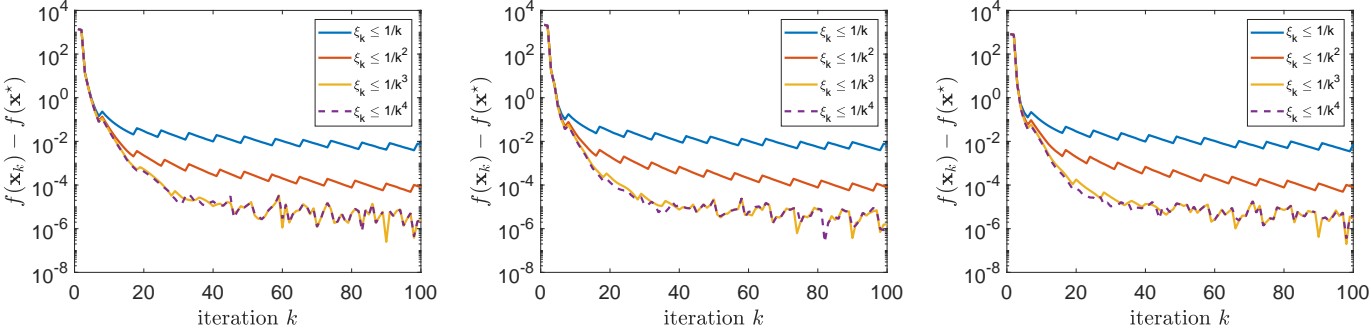

Figure 2: Results of linear coupling with inexact proximal operator for image deblurring with isotropic total variation. Objective function values v.s. number of iterations for different qualities of approximate solution of the proximal operator. From left to right: `Lena`, `Boat` and `Football`.

result. Specifically, the error sequences should decrease faster than $O(1/k^2)$ if the convergence rate of the algorithm is $O(1/k)$, for example, non-accelerated proximal method.

## 5.2 IMAGE DEBLURRING WITH ISOTROPIC TOTAL VARIATION

For the second experiment, we consider image deblurring with isotropic total variation regularization [Chambolle, 2004, Beck and Teboulle, 2009], which does not admit an analytical solution for the proximal operator [Chambolle and Pock, 2011, Beck and Teboulle, 2009]. For $\mathbf{X} \in \mathbb{R}^{m \times n}$, the discrete gradient operator is

$$(\nabla \mathbf{X})_{ij} = \begin{cases} (X_{ij} - X_{i+1,j}, X_{ij} - X_{i,j+1}) & \text{if } i < m, j < n \\ (0, X_{ij} - X_{i,j+1}) & \text{if } i = m, j < n \\ (X_{ij} - X_{i+1,j}, 0) & \text{if } i < m, j = n \end{cases}$$

Then, image deblurring with isotropic total variation regularization can be written as [Chambolle and Pock, 2016, Beck and Teboulle, 2009]

$$\min_{\mathbf{X} \in \mathcal{C}} \frac{1}{2} \|\mathcal{A}(\mathbf{X}) - \mathbf{B}\|_{\mathrm{F}}^2 + \lambda \sum_{i=1}^{m} \sum_{j=1}^{n} \|(\nabla \mathbf{X})_{ij}\|,$$

where $\mathcal{A} : \mathbb{R}^{m \times n} \to \mathbb{R}^{m \times n}$ is a linear operator representing some blurring processing, $\mathbf{B} \in \mathbb{R}^{m \times n}$ is the blurred and noisy image, and $\lambda > 0$ denotes a regularization parameter. Unlike the $\ell_1$ total variation, the proximal operator of isotropic total variation regularization does not admit an analytical solution. Same as before, we employ the BCD algorithm to compute an approximate solution.

We conducted this experiment on three images `Lena`, `Boat` and `Football` from MATLAB image processing toolbox. We first resize each image to $128 \times 128$ pixels. To obtain $\mathbf{B}$, the clean image was first blurred by a $5 \times 5$ kernel matrix: $\mathbf{S} = \frac{1}{25}\mathbf{I}_{5 \times 5}$, where $\mathbf{I}_{5 \times 5} \in \mathbb{R}^{5 \times 5}$ is an identify matrix, followed by additive Gaussian noise with zero mean and

standard deviation $10^{-1}$. The regularization parameter $\lambda$ was set to 0.1 for all three images.

Fig. 2 shows the objective function values versus the number of iterations of inexact linear coupling on `Lena`, `Boat` and `Football`. Similar trends as the first experiments are observed for this experiment except the performance between $\xi_k \leq 1/k^3$ and $\xi_k \leq 1/k^4$ are very close to each other.

In this experiment, we did not perform the experiments of ADMM due to it solves the subproblem associated with $\frac{1}{2}\|\mathcal{A}(\mathbf{X}) - \mathbf{B}\|_F^2$ is computationally expensive. Nevertheless, the experiment in Section 5.1 already clearly shows the advantages of linear coupling over ADMM.

## 6 CONCLUSION

Non-smooth regularizations have played irreplaceable roles in machine learning. However, many of them do not admit an analytical solution for proximal operator. As a variant of Nesterov's AGDs, the linear coupling can efficiently solve convex composite minimization with accelerated convergence rates if the proximal operator is exactly computed. In this work, we present a complete convergence analysis for linear coupling with inexact proximal operators. Our analysis suggests that inexact linear coupling still achieves the accelerated convergence rate if the error sequence of inexact proximal operator decreases at a sufficiently fast rate. More importantly, our theoretical results are better than previous works. We empirically verify our theoretical analysis by employing linear coupling with inexact operators to solve CUR-like factorization and image deblurring with isotropic total variation on different datasets.

### Acknowledgements

Qiang Zhou is supported by National Science Foundation of China under Grant 62106045 and Southeast University

Startup Fund (the Fundamental Research Funds for the Central Universities) under Grant 2242021R10096. Sinno J. Pan is supported by 2020 Microsoft Research Asia collaborative research grant.

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
