# OpenReview forum: "Convergence Analysis of Linear Coupling with Inexact Proximity Operator"
_auai.org/UAI/2022/Conference — UAI 2022 Poster_

### Official Review · Reviewer_DBAF · 2022-04-04

**Q2(1) Originality/Novelty:** 3
**Q2(2) Significance/Impact:** 3
**Q2(3) Correctness/Technical Quality:** 3
**Q2(6) Clarity Of Writing:** 3
**Q6 Overall Score:** 7
**Q8 Confidence In Your Score:** 3

**Q1 Summary And Contributions:**

This paper studies the convergence analysis of linear coupling with inexact proximal operator.

**Q2 Assessment Of The Paper:**

More detailed information regarding each of these aspects is given below:

**Q2(4) Quality Of Experiments (Optional):**

3: Good: The experimental evaluation is adequate, and the results convincingly support the main claims.

**Q2(5) Reproducibility:**

3: Good: Key resources (e.g., proofs, code, data) are available and key details (e.g., proofs, experimental setup) are sufficiently well-described for competent researchers to confidently reproduce the main results.

**Q3 Main Strengths:**

Please see Section 5.

**Q4 Main Weakness:**

Please see Section 5.

**Q5 Detailed Comments To The Authors:**

I am not an expert in this domain. In general, this paper is in good shape. Here are my comments.

1. This paper is easy to follow. Although it is a theoretical paper, the authors make the roadmap easy to follow, including the notation, preliminaries.
2. I believe Section 3 can be merged into Section 2, since it talks about linear coupling with inexact proximal operator.
3. For definitions, it is better to give a name to the definition.
4. The authors talk about ADMM. I think it is not very related to the research problem addressed in this paper. To my knowledge, ADMM  suffers from the convergence issues when determining the order of updating variables.
5. Following the above point, I think it is also unnecessary to incorporate ADMM in the experimental part.
6. There should be two lines indicating ADMM in Figure 1. I can only see one line. Maybe a different marker will help.
7. For the lines in Figure 1 and 2, the y-axis is log-scale. In this case, what is the threshold for algorithmic convergence?

**Q7 Justification For Your Score:**

This paper has a clear motivation. The presentation and organization is easy to follow. Although it is a theoretical paper, the authors also provide the experimental validation to make this paper complete.

**Q9 Complying With Reviewing Instructions:**

1: Yes.

---

### Official Review · Reviewer_YPPc · 2022-04-12

**Q2(1) Originality/Novelty:** 3
**Q2(2) Significance/Impact:** 2
**Q2(3) Correctness/Technical Quality:** 3
**Q2(6) Clarity Of Writing:** 4
**Q6 Overall Score:** 6
**Q8 Confidence In Your Score:** 3

**Q1 Summary And Contributions:**

In this paper, the authors study the convergence of linear coupling which accelerates first-order optimization algorithm. In particular, they focus on the case where the proximity operator cannot be computed exactly and must approximated. They find that if the approximation error decreases fast enough than an acceleration rate is recovered. Their theory also improves the classical case. They illustrate their analysis with examples from matrix factorization and image deblurring.

**Q2 Assessment Of The Paper:**

More detailed information regarding each of these aspects is given below:

**Q2(4) Quality Of Experiments (Optional):**

3: Good: The experimental evaluation is adequate, and the results convincingly support the main claims.

**Q2(5) Reproducibility:**

3: Good: Key resources (e.g., proofs, code, data) are available and key details (e.g., proofs, experimental setup) are sufficiently well-described for competent researchers to confidently reproduce the main results.

**Q3 Main Strengths:**

-The paper is very well-written and I enjoyed reading it. The algorithm is properly introduced and the main elements of the proofs are easy to follow via the Lemmas.

-Obtained bounds for accelerated algorithms under realistic approximate settings are interesting in view of applications where the proximal operator cannot be computed in closed forms.

-The proofs (I have thoroughly check the proofs in the convex case) are rigorous.

-The two experiments considered by the authors are non trivial (matrix factorization and image deblurring with TV regularization) and high dimensional. The empirical results validate the methodology

**Q4 Main Weakness:**

-I think the authors could have done a better job at describing the relations between their paper and the literature. I would have appreciated a section comparing explicitly the setting of [1] and the one considered in this paper. Also, what is precisely the advantage of the linear coupling?

-I have some concerns w.r.t. the experiments considered by the authors. Indeed all the plots are shown w.r.t. k but is it really the quantity to look at? More precisely, in order to reach \xi_k \leq 1/k^4 one needs a lot of evaluation to approximate the solutions with precision 1/k^4. What happens if we plot the evolution of the function w.r.t. the time (or the total number of call to one iteration of the proximity solver?). This remark asks the question of the relevance of studying accelerated version in an approximate context: is the cost of running longer the proximity solver worth the acceleration?

[1] Zhu, Orrechia -- Linear coupling: an ultimate unification of gradient and mirror descent

**Q5 Detailed Comments To The Authors:**

-I think that the authors should give a better discussion of the links between their work and [1]. Reading the introduction I found it quite hard to understand what was the exact novelty of the current paper (even though the comment on the \ell_2 norm was clear).

-Other papers study the properties of accelerated first order scheme with proximity operators such as [2] and the references therein.

-"bounded as following" (after Equation 26 in the supplementary) --> "bounded as follows"

-(still in the supplementary) "The last inequality obtained" --> "The last inequality is obtained"

-(still in the supplementary) I think that \leq (before "where the second inequality follows by" in the proof of Lemma 3) should be \geq.

-If I could follow the proof line by line in the supplementary, it would be better if the authors could provide a bit more of insight regarding the structure of the proof (in particular for Lemma 1/2/3).

[1] Schmidt, Le Roux, Bach -- Convergence rates of inexact proximal-gradient methods for convex optimization.

[2] Aujol, Dossal, Fort, Moulines -- Rates of Convergence of Perturbed FISTA-based algorithms

**Q7 Justification For Your Score:**

While I have some reservations w.r.t. the potential impact of perturbed accelerated first order methods (see my second comment in "Main weaknesses"), the results presented by the authors are new, rigorous and interesting. I appreciated that the authors illustrated their method with high dimensional examples.

**Q9 Complying With Reviewing Instructions:**

1: Yes.

---

### Official Review · Reviewer_WXAK · 2022-04-17

**Q2(1) Originality/Novelty:** 3
**Q2(2) Significance/Impact:** 3
**Q2(3) Correctness/Technical Quality:** 3
**Q2(6) Clarity Of Writing:** 3
**Q6 Overall Score:** 7
**Q8 Confidence In Your Score:** 2

**Q1 Summary And Contributions:**

The paper provides a convergence analysis of linear coupling when the proximal operator can only be computed approximately. This scenario occurs, for example, when a norm other than L2 is used. The theoretical convergence rates are better than previously known ones, and the algorithm's behavior is verified in experiments.

**Q2 Assessment Of The Paper:**

More detailed information regarding each of these aspects is given below:

**Q2(4) Quality Of Experiments (Optional):**

3: Good: The experimental evaluation is adequate, and the results convincingly support the main claims.

**Q2(5) Reproducibility:**

3: Good: Key resources (e.g., proofs, code, data) are available and key details (e.g., proofs, experimental setup) are sufficiently well-described for competent researchers to confidently reproduce the main results.

**Q3 Main Strengths:**

- Theoretical derivation of convergence rates
- Experimental validation of the results

**Q4 Main Weakness:**

- All proofs are relegated to the appendix

**Q5 Detailed Comments To The Authors:**

The paper's topic is not in my area of expertise, and as such, it is possible that aspects that are unclear to me or I found challenging to follow are a non-issue to people working more closely in this field. This also means that my ability to assess the novelty and impact of the work is limited.

Overall I found the paper easy to read, and while I did not verify the proofs. The order in which the theoretical aspects are introduced and built on top of each other made sense. There are sentences throughout the paper that seem grammatically odd, which another pass should address. This is not a real issue, just something that stood out a bit while reading the paper. While I appreciate the exposition in the introduction, which explains the need for the proposed work, it felt a bit repetitive/long.

A few questions that I had while reading the paper:

- The convergence analysis assumes that psi(x) is strongly convex and rho-smooth with regards to the norm. Is this limiting, and if so, in which manner?
- In Section 4.3, it is assumed that the norm is the L2 norm. This is different from the previous sections (if I understood correctly). Why is this additional restriction added, and what is its impact? Could the same properties be obtained for other norms?

For a theory-focused paper, it would be nice if not all proofs were relegated to an appendix. I do not expect every single proof to be present in full. However, it would be great for the core theorem/conclusion if at least a proof sketch would be provided.

While brief and showcasing exemplary results only, the experiments are well presented and demonstrate that the theoretical aspects hold. One aspect that would be nice to talk about is how realistic the required approximation qualities are, i.e. how hard it is to achieve the required precision. This likely differs between problem setups, but a rough idea would be interesting. One minor improvement that could be made to the plots is to move the legend underneath the plots and have the title refer to the dataset being used.

**Q7 Justification For Your Score:**

As the area of the paper is outside my area of expertise, the score has to be taken with a grain of salt. For the most part, the paper is easy to read and can follow. The results appear promising, and the experiments do an adequate job for a theory-heavy paper. If the novelty is there, I believe that this paper would positively impact the theoretical understanding of linear coupling methods.

**Q9 Complying With Reviewing Instructions:**

1: Yes.

---

### Decision · Program_Chairs · 2022-05-15

**Decision:**

Accept (Poster)

**Comment:**

Meta Review: The authors analyze the convergence rate of linear coupling for optimizing a composite convex loss consisting of a smooth and non-smooth component, such as L1-regularized learning, in the setting of an inexactly optimized proximal update.  They develop a convergence rate as a function of a sequence of proximal suboptimality bounds, $\xi_k$, and show that their resulting bounds are tighter than those of prior work.

Although the contribution of the work is mainly theoretical, reviewers did have some concerns about the focus and scope of the experimental validation.  However, the author responses were helpful in understanding their choices and reported results.